# Multicentre, randomised controlled trial to investigate the effects of parental touch on relieving acute procedural pain in neonates (Petal)

Maria M Cobo,[1,2] Fiona Moultrie,[1] Annalisa G V Hauck ,[1] Daniel Crankshaw,[1] Vaneesha Monk,[1] Caroline Hartley,[1] Ria Evans Fry,[1] Shellie Robinson,[1] Marianne van der Vaart,[1] Luke Baxter ,[1] Eleri Adams,[3] Ravi Poorun,[4,5] Aomesh Bhatt,[1] Rebeccah Slater[1]

[1]Department of Paediatrics, University of Oxford, Oxford, UK
[2]Colegio de Ciencias Biologicas y Ambientales, Universidad San Francisco de Quito USFQ, Quito, Ecuador
[3]Newborn Care Unit, John Radcliffe Hospital, Oxford University Hospitals NHS Foundation Trust, Oxford, UK
[4]Children's Services, Royal Devon University Healthcare NHS Foundation Trust, Exeter, UK
[5]College of Medicine & Health, University of Exeter, Exeter, UK

**Correspondence to**
Rebeccah Slater;
rebeccah.slater@paediatrics.ox.ac.uk

## ABSTRACT

**Introduction** Newborn infants routinely undergo minor painful procedures as part of postnatal care, with infants born sick or premature requiring a greater number of procedures. As pain in early life can have long-term neurodevelopmental consequences and lead to parental anxiety and future avoidance of interventions, effective pain management is essential. Non-pharmacological comfort measures such as breastfeeding, swaddling and sweet solutions are inconsistently implemented and are not always practical or effective in reducing the transmission of noxious input to the brain. Stroking of the skin can activate C-tactile fibres and reduce pain, and therefore could provide a simple and safe parent-led intervention for the management of pain. The trial aim is to determine whether parental touch prior to a painful clinical procedure provides effective pain relief in neonates.

**Methods and analysis** This is a multicentre randomised controlled trial. A total of 112 neonates born at 35 weeks' gestation or more requiring a blood test in the first week of life will be recruited and randomised to receive parental stroking either preprocedure or postprocedure. We will record brain activity (EEG), cardiac and respiratory dynamics, oxygen saturation and facial expression to provide proxy pain outcome measures. The primary outcome will be the reduction of noxious-evoked brain activity in response to a heel lance. Secondary outcomes will be a reduction in clinical pain scores (Premature Infant Pain Profile-Revised), postprocedural tachycardia and parental anxiety.

**Ethics and dissemination** The study has been approved by the London—South East Research Ethics Committee (ref: 21/LO/0523). The results will be widely disseminated through peer-reviewed publications, international conferences and via our partner neonatal charities Bliss and Supporting the Sick Newborn And their Parents (SSNAP). If the parental tactile intervention is effective, recommendations will be submitted via the National Health Service clinical guideline adoption process.

**Study status** Commenced September 2021.

**Trial registration number** NCT04901611; 14 135 962.

## STRENGTHS AND LIMITATIONS OF THIS STUDY

⇒ Petal is a randomised controlled trial investigating whether noxious-evoked brain activity is reduced by preprocedural parental stroking.
⇒ The trial is based on published evidence from two mechanistic studies which show a reduction in noxious-evoked brain activity during a heel lance or experimental stimuli in neonates whose skin was brushed by the experimenter prior to the procedure.
⇒ This trial investigates stroking as a simple, free, low-risk, non-pharmacological pain-relieving intervention delivered by parents to their newborn infants in the first week of life.
⇒ The trial employs multiple proxy measures to determine the impact of the stroking intervention on neonatal pain and investigates the impact of the intervention on parental anxiety and distress.
⇒ While investigators cannot be blinded to the group allocation at the time of the study, this limitation is mitigated by ensuring that participants and investigators involved in all other aspects of the trial, including data analysis, are blinded.

## INTRODUCTION
### Background

Newborn infants undergo painful procedures as part of routine neonatal care. Sick or premature infants experience an average of 10 painful procedures per day as part of life-sustaining treatment.[1] It is recognised that repetitive exposure to pain in early life can cause short-term physiological instability as well as long-term neurodevelopmental consequences such as reduced growth, altered structural and functional brain development and reduced school-age academic performance.[2] Furthermore, repeatedly witnessing their infant in pain can have a significant negative impact on the emotional and psychological well-being of parents.[3–5] Effective pain

management is therefore essential in neonatal care. However, measuring pain in this non-verbal patient population is challenging, and few safe and effective analgesics have been tested and approved for use in infants. Non-pharmacological strategies have been introduced and promoted over the last few decades for the management of acute procedural pain. Sweet-taste solutions such as sucrose are effective in relieving behavioural responses following minor painful procedures,[6] but do not reduce noxious input to the brain.[7] This has caused concern that this intervention may not mitigate the long-term consequences of early life pain, and furthermore, it may have long-term neurodevelopmental effects with repeated use.[8–10] Breastfeeding also reduces behavioural and physiological responses to pain in full-term infants undergoing heel lancing, intramuscular injection and venepuncture.[11] However, this strategy can be challenging for new mothers and is not always practical to implement in premature and critically ill infants or in mothers with transmissible infections. Other comfort measures include swaddling and facilitated tucking of infants, which, although useful, are less effective in reducing pain.[12] While many studies have reported the potential pain-relieving effects of tactile interventions such as skin-to-skin care[13] and massage[14–21] in the context of minor painful procedures, these non-pharmacological interventions are scarcely used in maternity and neonatal units[22 23 22] and the mechanisms underpinning their effectiveness are still being established. Despite guidelines recommending the use of non-pharmacological interventions for pain relief, uptake of these practices remains poor and inconsistent.[23 24]

### Measuring pain in infants

The assessment of pain and analgesia in infants primarily relies on measuring changes in infant behaviour. One of the most common validated clinical pain tools is the Premature Infant Pain Profile (original PIPP, revised PIPP-R).[25 26] While subjective evaluations of behavioural responses are a gold standard for the clinical assessment of neonatal pain, electrophysiology-based methods have more recently been developed to identify a pattern of noxious-evoked brain activity.[27–29] This objective and quantifiable neurophysiological measure has been previously used in pilot studies[30 31] and as the primary outcome measure in randomised clinical trials published in *The Lancet,* assessing the analgesic efficacy of sucrose[7] and morphine.[32] Noxious-evoked brain activity has specifically been well characterised in response to heel lancing,[27–29 33] a clinical procedure which is frequently performed in neonates for blood collection, and will be used as the primary outcome of the Petal trial to investigate the efficacy of preprocedural parental stroking.

### Rationale

Maternal touch behaviours are instinctive, evolutionarily conserved among mammals.[34] Previous studies suggest that there may also be a potential relationship between enhanced maternal touch and infant growth and development.[35 36] Stroking, by repeatedly applying gentle pressure to the skin, can activate C-tactile (CT) fibres, a subclass of slow-conducting unmyelinated sensory neurons, mostly found in hairy skin.[37–39] These fibres project to brain regions associated with affective processing such as the insular cortex, prefrontal cortex, superior temporal sulcus and cingulate cortex[40–44] and are thought to have evolved to promote affiliative behaviours and social touch.[45–48] CT-fibres are optimally activated by stroking at a velocity of 3 cm/s (optimal range 1–10 cm/s),[49–51] and studies in adults of gentle brushing or stroking paradigms at this optimal velocity have demonstrated a reduction in pain ratings[52 53] and noxious-evoked brain activity.[53] CT-optimal stimulation therefore could provide a natural and safe pain-relieving intervention.

We previously conducted a small prospective cohort study of preprocedural stroking for pain relief in neonates, in which we demonstrated that CT-optimal stroking (at 3 cm/s) prior to an experimental noxious stimulus or clinical heel lance significantly reduced noxious-evoked brain activity in term neonates compared with no touch intervention.[30] We replicated this study in an independent sample of term neonates and showed consistent results and a similar effect size in the group receiving the stroking intervention.[31] However, in both of these studies, stroking was delivered by the researcher using a soft experimental brush with a known force. Although the studies did not identify a significant effect of the intervention on a clinical pain score, they were notably not powered to investigate this. Considering CT-optimal stroking is a natural maternal behaviour[54 55] and evidence suggesting that CT-fibres respond optimally to touch at human skin temperature,[56] hands-on parental stroking has the potential to provide even greater benefit than CT-optimal brushstrokes. Pilot work further suggests that stroking a neonate has similar efficacy to researcher-led experimental brushing (unpublished).

### Aim and objectives

In the Petal trial, we aim to determine whether parental stroking prior to a common painful clinical procedure (heel lancing) provides effective analgesia in neonates. The primary outcome will be the reduction of noxious-evoked brain activity during a heel lance. Secondary outcomes will be a reduction in clinical pain scores, postprocedural tachycardia and parental anxiety (table 1). Exploratory outcomes will investigate changes in brain activity during the intervention, as well as effects on physiological recovery postprocedure (using heart rate and respiratory dynamics) and further explore parental anxiety, distress, and attitudes to research.

## METHODS AND ANALYSIS
### Trial description

This is a multicentre randomised controlled interventional trial, with two research sites (John Radcliffe Hospital, Oxford, and Royal Devon and Exeter Hospital, Devon, UK). The parents of eligible neonates satisfying

**Table 1** Objectives and outcome measures

| Objectives | Outcome measures |
|---|---|
| **Primary objective** | **Primary outcome measure** |
| 1. To test whether parental touch prior to the clinical procedure reduces noxious-evoked brain activity following a heel lance. | 1. Magnitude of noxious-evoked brain activity following a heel lance (EEG data recorded in the 1000 ms period following each heel lance). |
| **Secondary objectives** | **Secondary outcome measures** |
| 1. To test whether parental touch prior to the clinical procedure reduces clinical pain scores (PIPP-R) during the 30 s period after the heel lance. | 1. PIPP-R score during the 30 s period after the heel lance. |
| 2. To test whether parental touch prior to the clinical procedure reduces incidence of postprocedural tachycardia following a heel lance. | 2. Percentage of neonates who develop tachycardia in the 30 s post heel lance. |
| 3. To test whether parental touch prior to the clinical procedure reduces parental anxiety, compared with postprocedural touch. | 3. Difference in STAI-S scores preprocedure and postprocedure. |
| **Exploratory objectives** | **Exploratory outcome measures** |
| 1. To explore how parental touch impacts background brain activity. | 1. Changes in brain activity during the touch intervention. |
| 2. To explore whether parental touch prior to the clinical procedure reduces the duration of time for heart rate to return to baseline after a heel lance. | 2. Time taken for heart rate to return to baseline post heel lance. |
| 3. To explore how parental touch prior to the clinical procedure affects respiratory stability. | 3. Postprocedural respiratory dynamics and incidence of apnoea. |
| 4. To explore parental anxiety and distress, and their experience of the trial and infant research. | 4. Scores for individual parameters from the STAI-T and STAI-S; four-point distress questionnaire score; responses to survey about participation in Petal and infant research. |

EEG, Electroencephalography; PIPP-R, Premature Infant Pain Profile-Revised; STAI-S, State-Trait Anxiety Inventory-State; STAI-T, State-Trait Anxiety Inventory-Trait.

inclusion criteria (figure 1) will be approached by a member of the research team. Parental written informed consent will be taken and neonates will be electronically randomised to receive parental stroking either prior to or after a clinically required heel lance. Patient information leaflets and consent forms are available as online supplemental file 1. A unique study ID will be assigned to each individual participant. The randomisation programme will use a minimisation algorithm to ensure approximate balance between the groups with respect to gestational age at birth, postnatal age at time of randomisation, sex, the indication for blood sampling and research site. The users of the system will be blind to the next allocation.

Each neonate will be studied on a single test occasion lasting approximately 1 hour (figure 2) and will not require further follow-up. A parent will first complete the State-Trait Anxiety Inventory-Trait (STAI-T) and State-Trait Anxiety Inventory-State (STAI-S) questionnaires, which will be administered verbally by a researcher. This will allow assessment of both trait anxiety and state anxiety prior to the commencement of the stroking intervention or blood test. At least 30 min prior to the heel lance, the research team will set up physiological monitoring including ECG and pulse oximetry for continuous recording of baseline cardiorespiratory stability. Electroencephalography (EEG) electrodes will then be sited to allow continuous monitoring of baseline brain activity for at least 10 min prior to the clinical procedure. A control

heel lance will then be performed followed by the clinical heel lance.

The control heel lance is a non-noxious sham procedure whereby the lancet is placed against the participant's foot rotated at 90°, preventing release of the blade into the foot. This procedure simulates the tactile and auditory aspects of the blood sampling experience without the noxious input. Brain activity, physiology and facial expression (video) will be recorded for both the control heel lance and clinical heel lance to allow assessment of outcome measures including noxious-evoked brain activity, PIPP-R scores, tachycardia and respiratory dynamics (table 1). The heel lance and control stimulus will be linked electronically to the recording equipment as described in previous studies,[7 27 28] providing precise timing of when the heel lance occurs. In the event of the neonate requiring multiple heel lances, data will only be included from the first heel lance (conditional on data quality). Video recording of the face will commence approximately 30 s prior to the control heel lance and end at least 30 s after the clinical heel lance to allow PIPP-R scoring. For neonates randomised to receive preprocedural stroking, the parent will be instructed to begin stroking down the infant's leg immediately prior to the clinical heel lance, with the aid of an animated visual cue to help maintain a velocity of 3 cm/s and a duration of 10 s. After the heel lance, blood collection will be delayed for 30 s to allow PIPP-R scoring. For neonates randomised

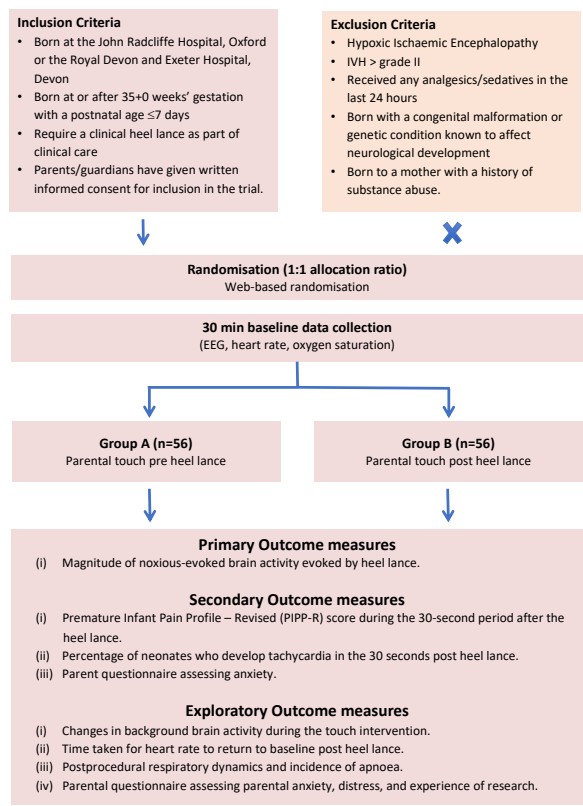

**Inclusion Criteria**
- Born at the John Radcliffe Hospital, Oxford or the Royal Devon and Exeter Hospital, Devon
- Born at or after 35+0 weeks' gestation with a postnatal age ≤7 days
- Require a clinical heel lance as part of clinical care
- Parents/guardians have given written informed consent for inclusion in the trial.

**Exclusion Criteria**
- Hypoxic Ischaemic Encephalopathy
- IVH > grade II
- Received any analgesics/sedatives in the last 24 hours
- Born with a congenital malformation or genetic condition known to affect neurological development
- Born to a mother with a history of substance abuse.

**Randomisation (1:1 allocation ratio)**
Web-based randomisation

**30 min baseline data collection**
(EEG, heart rate, oxygen saturation)

**Group A (n=56)**
Parental touch pre heel lance

**Group B (n=56)**
Parental touch post heel lance

**Primary Outcome measures**
(i) Magnitude of noxious-evoked brain activity evoked by heel lance.

**Secondary Outcome measures**
(i) Premature Infant Pain Profile – Revised (PIPP-R) score during the 30-second period after the heel lance.
(ii) Percentage of neonates who develop tachycardia in the 30 seconds post heel lance.
(iii) Parent questionnaire assessing anxiety.

**Exploratory Outcome measures**
(i) Changes in background brain activity during the touch intervention.
(ii) Time taken for heart rate to return to baseline post heel lance.
(iii) Postprocedural respiratory dynamics and incidence of apnoea.
(iv) Parental questionnaire assessing parental anxiety, distress, and experience of research.

**Figure 1** Trial flowchart. IVH, intraventricular haemorrhage.

to receive postprocedural stroking, the parent will be instructed to begin stroking down the infant's leg after the start of blood collection, when deemed appropriate by the clinician performing the heel lance in order to ensure that blood collection is not disrupted. Parents will be guided by an animated visual cue. A researcher will then verbally administer the STAI-S and four-point distress questionnaire after the procedure is completed. Physiological monitoring will continue for 30 min and EEG monitoring for at least 10 min to allow investigation of postprocedural cardiorespiratory dynamics and brain activity as exploratory outcomes of the trial. Finally, the parents will be invited to complete an anonymous survey of their experience and views on research after completion of the study. This study protocol follows the Standard Protocol Items: Recommendations for Interventional Trials guidelines (online supplemental file 2).[57]

## Intervention

The parental touch intervention will involve one parent stroking the infant's leg for 10 s. The duration of the intervention is consistent with previous studies.[30 31 52 58] A member of the research team will inform parents of their randomised allocation (either stroking pre heel lance or post heel lance) at the start of the test occasion. They will explain and demonstrate how to administer the intervention using their whole hand, stroking in one direction down towards the foot. The infant will lay in a

cot during the intervention and procedure. During the demonstration and test occasion, PsychoPy software[59] will be used to provide a visual cue on a computer screen to guide a consistent stroking speed of 3 cm/s for 10 s. During the study, all neonates will receive comfort care in accordance with the local practice guidelines. These measures include swaddling the infants and providing non-nutritive sucking.

## Recording techniques

### Electroencephalography (EEG)

Electrophysiological activity will be acquired with the SynAmps RT 64-channel headbox and amplifiers (Compumedics Neuroscan) or with the Compumedics Grael V2 EEG system, with a bandwidth from DC: 400 Hz and a sampling rate of 2000 or 2048 Hz. Data recorded at 2048 Hz will be downsampled to 2000 Hz prior to further processing. CURRYscan7 or CURRYscan8 neuroimaging suite (Compumedics Neuroscan) will be used to record the activity. All equipment will conform to the electrical safety standard for medical devices, IEC 60601-1. Eight EEG recording electrodes will be positioned on the scalp at Cz, CPz, C3, C4, FCz, T3, T4 and Oz according to the modified international 10–20 System. Reference and ground electrodes will be placed at Fz and Fpz, respectively. EEG conductive paste will be used to optimise contact with the scalp. All impedances will be reduced to approximately 5 kΩ by rubbing the skin with EEG preparation gel prior to electrode placement. An ECG electrode will be placed on the left clavicle to record heart rate.

### Physiological monitoring (ECG and pulse oximetry)

Heart rate, respiratory rate and oxygen saturation will be recorded continuously throughout the study period (approximately 1 hour) using ECG and pulse oximetry. Heart rate and oxygen saturation data will be used to calculate the clinical pain scores following the heel lance and control stimulus and to assess clinical stability across the test occasion.

### Video recording

Video recording will be used to measure behavioural responses that is, changes in facial expression during the control stimulus and clinically required heel lance. A synchronised LED flash will be activated by the researcher simultaneously with each stimulation as a marker for the time of stimulation.

### Parental questionnaire

The parent administering the intervention will be asked to complete a short series of validated electronic questionnaires assessing anxiety and distress at the start and end of the test occasion (table 2). The researcher will record the responses to the STAI-T, STAI-S and distress questionnaire in an electronic Case Report Form. The electronic device will then be presented to the parent to independently complete a short survey about trial participation and their research experience. The survey will be

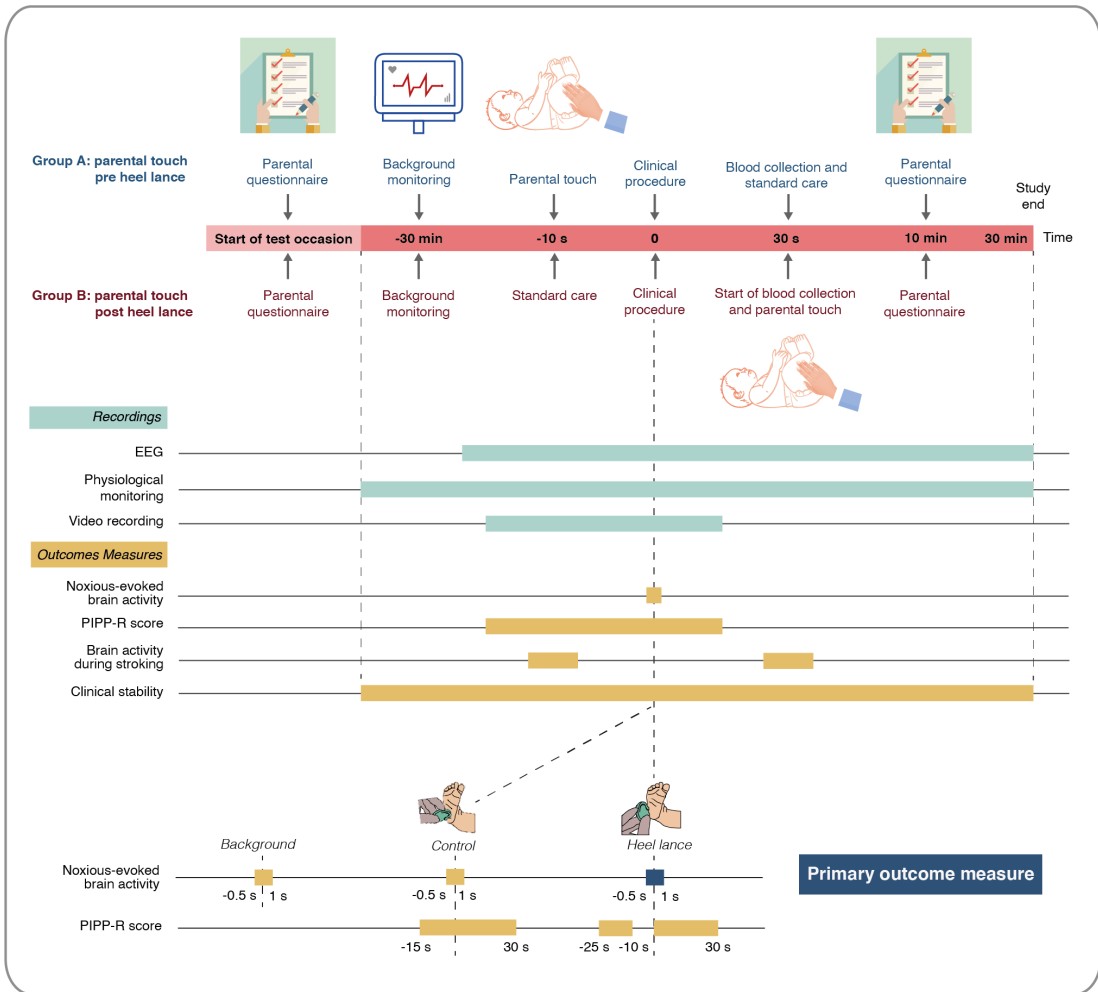

**Figure 2** Trial procedures. EEG, electroencephalography; PIPP-R, Premature Infant Pain Profile-Revised.

completed anonymously, and responses will be stored by trial arm with no link to study IDs.

### State-Trait Anxiety Inventory (STAI)
The State-Trait Anxiety Inventory (STAI) is the gold standard assessment for state anxiety.[60] It is well validated, publicly available and has a trait (STAI-T) version consisting of 20 statements exploring general feelings of anxiety, and a state version (STAI-S) consisting of 20 statements exploring anxiety levels at a particular point in time. Each question is rated on a four-point scale. The range of possible scores for the STAI varies from a minimum score of 20 to a maximum score of 80 on both the STAI-T and STAI-S subscales.

### Four-point distress questionnaire
Parents will be asked four questions related to their emotions during the clinical heel lance procedure.[61 62]

| Table 2 | Trial parental questionnaires | | |
|---|---|---|---|
| **Questionnaire section** | **Topic** | **Timing of administration** | **Questionnaire administrator** |
| 20-point State-Trait Anxiety Inventory (STAI)-T | Trait anxiety | Start of test occasion | Administered verbally by researcher |
| 20-point State-Trait Anxiety Inventory (STAI)-S | State anxiety pre heel lance State anxiety post heel lance | Start of test occasion After the procedure and intervention are completed | Administered verbally by researcher Administered verbally by researcher |
| Four-point distress questionnaire | Emotional constructs experienced at time of the clinical heel lance | After the procedure and intervention are completed | Administered verbally by researcher |
| Anonymous survey | Views on the trial and infant research | End of test occasion | Completed by parent |

Each of the four emotional constructs (worried, upset, anxious and sad) will be rated on an 11-point scale ranging from 'not at all' (0) to 'extremely' (10). A total score between 0 and 40 will be calculated, where higher scores are indicative of greater parental distress. This score is frequently used in research to evaluate parent/child interactions during painful procedures.[61–63]

## Outcome measures

### Noxious-evoked brain activity

An EEG template that reflects the noxious-evoked brain activity in neonates has previously been defined using principal component analysis, validated in independent data sets[29] and used in clinical studies and a clinical trial.[32] This template will be projected onto the EEG data recorded in the 1000 ms period following each heel lance and control heel lance stimulus and the relative weight of the component calculated for each neonate. A greater weight indicates a stronger noxious-evoked response. While the brain activity characterised is directly related to noxious input, it does not reflect all noxious-evoked activity across the brain or all aspects of the pain experience. The response to the non-noxious control heel lance stimulus is being recorded to confirm that it significantly differs from the brain activity evoked by a noxious heel lance. This forms an important data quality control check.[27]

### PIPP-R score

Clinical pain scores will be evaluated using the validated Premature Infant Pain Profile-Revised,[26] which is a composite multimodal measure encompassing behavioural, physiological and contextual indicators of the pain response. It allows for different aspects of the infant pain experience to be captured and has been widely used as the primary outcome measure for infant pain in many clinical trials.[64–66] The PIPP-R score will be calculated for the control heel lance and the clinical heel lance procedure. Heart rate, oxygen saturation and facial expression will be recorded in the 15 s period before and 30 s period after each of the procedures.[25 26] The 15 s period before the heel lance will be recorded immediately prior to the stroking intervention. Videos of the infant's facial expressions will be scored offline using the PIPP-R facial coding system. Changes in heart rate and oxygen saturation will be recorded with ECG and pulse oximeter and used to calculate the PIPP-R score. For each participant, PIPP-R scores will be assessed by investigators blinded to the study arm. A second investigator (blinded to the trial arm) will recalculate 20% of the PIPP-R scores to measure inter-rater reliability.

### Clinical stability

Clinical stability will be assessed in the 30 min periods before and after the heel lance. The percentage of neonates who develop postprocedural tachycardia in the 30 s postheel lance will be a secondary outcome measure of the trial. Tachycardia will be defined as a heart rate >160 beats per minute as per Advanced Paediatric Life Support guidelines, reflecting heart rate values >90th centile for newborn infants in the first week of life.[67 68] Exploratory outcome measures will also include the time taken for the heart rate to return to baseline values post heel lance and respiratory rate variability in the 30 min prior and post heel lance (including incidence of apnoea). An episode of apnoea will be defined as the cessation of breathing for at least 20 s.[69]

### Parental experience

Parental anxiety will be quantified using the outcomes of the STAI-T and STAI-S questionnaires. Parental distress will be quantified using the four-point distress score. The anonymous parent survey will assess the parental experience of the trial and parental views on taking part in the trial.

## Statistics and analysis

### Analysis of outcome measures

Data preprocessing and statistical analysis will be performed blind to treatment allocation. The analysis and presentation of results will follow the most up-to-date recommendations of the Consolidated Standards of Reporting Trials group (CONSORT).[70] All comparative analyses will be performed using MatlabR2020a or an updated version. The primary results will be presented unadjusted. To perform sensitivity analysis, the minimisation variables will be used to make statistical adjustments to the primary analysis and the sensitivity analysis results will be presented as secondary results. A full statistical analysis plan will be finalised before any comparative analysis of outcome measures is performed.

### Significance levels

For the analysis of the primary outcome measure, a p-value of 0.05 (two-sided 5% significance level) will be used to indicate statistical significance. Significance levels for secondary outcomes (excluding the sensitivity analysis) will be corrected for multiple comparisons and the method will be specified in the analysis plan. Two-sided statistical tests and corresponding p-values will be presented throughout; however, for the purposes of interpretation of results, CIs will dominate, rather than p-values.

### Primary

#### *Noxious-evoked brain activity*

The magnitude of noxious-evoked brain activity will be compared between the two groups using a parametric two-sample t-test if the residuals are normally distributed. If the residuals are non-normally distributed, a Wilcoxon rank-sum test will be used. If appropriate, and depending on the distribution of residuals and the test used, the mean and SD or the median and IQR (or entire range, whichever is appropriate) will be presented for each group and the unadjusted mean or median difference between groups with a 95% CI.

## Secondary

### PIPP-R score

PIPP-R scores (during the 30s period after heel lance) in the two groups will be compared using a two-sample t-test if the residuals are normally distributed. If the residuals are non-normally distributed, a Wilcoxon rank-sum test will be used. If appropriate, and depending on the distribution of residuals and the test used, the mean and SD or the median and IQR (or entire range, whichever is appropriate) will be presented for each group and the unadjusted mean or median difference between groups with a 95% CI.

### Clinical stability (tachycardia)

The tachycardia outcome per infant will be dichotomous (i.e. 'yes/no' per infant). The percentage of infants experiencing tachycardia will be compared between the two groups using a logistic regression. We will report the proportion of tachycardia for each group as well as the difference in proportions between groups.

### Parental anxiety

The difference in STAI-S scores before and after the heel lance will be compared between the two groups using a two-sample t-test if the residuals are normally distributed. If the residuals are non-normally distributed, a Wilcoxon rank-sum test will be used. If appropriate, and depending on the distribution of residuals and the test used, the mean and SD or the median and IQR (or entire range, whichever is appropriate) will be presented for each group and the unadjusted mean or median difference between groups with a 95% CI.

### Exploratory

Exploratory analyses will be conducted to investigate (i) the effects of parental touch on background brain activity, (ii) whether preprocedural parental touch reduces the duration of time for heart rate to return to baseline, (iii) the effect of preprocedural parental touch on respiratory rate variability, respiratory dynamics and the incidence of apnoea and (iv) the parental experience of the procedure and involvement in research.

## Sample size determination

### Power calculation

The assumptions for these calculations are based on data from mechanistic studies investigating the effect of (experimenter-led) soft brushing of the skin at CT-optimal rate on the response to an experimental noxious stimulus or clinical heel lance in term neonates.[30 31] The mean (SD) brain activity evoked by heel lancing in the control group is estimated to be 1.07 (0.66). A 40% reduction in the intervention group is considered to be clinically significant and realistic from other studies.[30 31 53] With 90% power and a two-sided 5% significance level, to observe a 40% reduction in brain activity with a two-sample t-test, a sample size of 102 would be required. Allowing for 10% loss, due to technical difficulties or other clinical issues, this increases to 112.

## Missing data

Missing data may occur in our trial due to equipment failure, EEG artefacts or clinical issues resulting in withdrawal post randomisation. If missing data exists, we expect it will occur at random, and collected data will be representative of the population. To account for potential missing data, we have inflated our sample size by 10%. The analysis will be conducted using the available data.

## Ethics and dissemination

The trial has been approved by the London South East Research Ethics Committee (ref: 21/LO/0523) and will be conducted in accordance with Good Clinical Practice and the Declaration of Helsinki. EEG is a safe tool used routinely in clinical practice and research to measure brain activity. Surface electrodes are used and temporarily fixed without glue. All heel lances performed during the trial will have been requested by the clinical team responsible for the infant's medical care. No extra blood tests or noxious procedures will be performed for the purpose of the study. Every effort will be made to minimise inconvenience and prevent disruption of clinical care. There are no expected serious adverse events (SAE) for this trial. Any SAEs identified will be reported to the CI within 24 hours and they will report any unexpected SAEs deemed related to the trial to the REC and Sponsor in accordance with REC/HRA guidance.

Parent(s) may withdraw their neonate from the trial at any time and they are not obliged to give a reason. If parents choose to withdraw their child after the study has begun, they will be asked whether data already collected may be retained and used for the purposes of the trial. Parents will be made aware that this decision has no impact on any aspects of their infant's continuing care. The attending clinician may also withdraw the neonate from the trial if they consider this to be in their best interest. If any of the exclusion criteria manifest prior to data collection, the participant will be withdrawn.

The results of the study will be disseminated to the scientific and wider community through peer-reviewed publications and national and international meetings and conferences, via the charities Supporting the Sick Newborn And their Parents (SSNAP) and Bliss, and through the National Health S clinical guideline adoption process.

## Patient and public involvement (PPI)

A PPI representative will be included in the extended PMG group and invited to join specific PMG meetings to discuss trial progress and developments. Bliss: for babies born premature or sick is a national UK neonatal charity, which is partly funding the trial. They will receive regular trial progress reports and promote the trial across their various channels, and disseminate the results. The research team will also work closely with the onsite local Oxford charity SSNAP during the design, conduct and dissemination of the trial. SSNAP have reviewed all parent-facing materials, will review manuscripts reporting

results and will be involved in disseminating results to the public.

## DISCUSSION

All newborn infants are exposed to clinically necessary painful procedures. Even healthy neonates on postnatal wards can require repeated painful procedures beyond routine Newborn Screening, such as blood tests for glucose monitoring or jaundice, which can be distressing for both neonates and their parents. In the UK, more than 100 000 newborn infants receive neonatal care every year as a result of prematurity or illness,[71] which, for some, can entail weeks to months of hospitalisation and procedures. As such, improving the management of pain is recognised as a top neonatal UK research priority[72] and a major concern among parents and neonatal nurses.[73]

Poor management of neonatal pain can have a significant negative impact on parents. Mothers of hospitalised infants report feeling emotionally and psychologically traumatised due to having to allow their infants to undergo clinically necessary painful procedures, and due to feelings of helplessness from being unable to protect or comfort their child.[3–5] Actively involving parents in care relieves parental distress[74] and increases the likelihood that infants receive treatment for pain.[1 5 75] Infant massage, a tactile comfort measure which involves patterns of stroking, has been shown to improve mother–infant bonding and improve postnatal depression,[76] a condition afflicting at least one in ten UK mothers in the first-year postpartum.[77] Furthermore, maternal stroking of infants in general has been shown to moderate the behavioural and physiological effects of maternal depression on infants.[78] Promoting the natural tactile behaviour of stroking to provide evidence-based pain-relief would therefore be beneficial to both mothers and infants.

Anxiety about pain is increasingly recognised as a key factor in parental refusal for procedures such as vitamin-K intramuscular injections at birth[79] and immunisations.[80–82] Avoidance of key interventions in early life could have drastic consequences for child health and this issue must be addressed. Indeed, parental anxiety and attitudes during painful procedures can also impact neonatal distress and subsequent pain experience during clinical procedures in later life.[83] Parental anxiety regarding pain could be alleviated by empowering parents to provide safe and effective pain relief for their child. Unlike other non-pharmacological interventions, this strategy could be broadly implemented regardless of feeding status of the infant or availability of a product like sucrose, in hospital as well as the community, and across high and low resource clinical settings.

CT-fibres likely provide the neurobiological mechanism underlying the benefits of tactile stimulation in early life. Studies have revealed that mothers instinctively stroke their infants at a CT-optimal rate[54 55] and that this tactile stimulation is beneficial. CT-optimal touch significantly decreases resting heart rates in infants aged 1–4 months[84] and 9 months,[58] as well as in premature infants (28–36 weeks' gestation).[85] Recent studies have also investigated the neurological correlates of CT-optimal touch in early life. In 2-month-old infants, CT-optimal touch produces greater activation of the insular cortex compared with CT non-optimal touch.[84] Similarly, in term infants CT-optimal stroking with a soft brush produces activation of the primary somatosensory and posterior insular cortices,[86] suggesting that the neonatal brain is sensitive to the somatosensory and socio-affective effects of CT-optimal stroking.

The Petal trial is based on clear mechanistic evidence from preliminary cohort studies and is, as such, adequately powered to address the clinical question. It employs a range of multimodal outcomes, including electrophysiological, behavioural and cardiorespiratory measures, to cover the many aspects of pain experience, and seeks to investigate the benefits of the intervention to both neonates and their parents. Blinding of outcome assessment is being performed to ensure the integrity of the trial as it is not possible to blind the researchers at the time of study due to the nature of the intervention. Although parents instinctively stroke at the optimal velocity to stimulate CT-fibres,[54 55] consistency of the intervention is standardised across the trial by providing an animated visual aid for parents to follow. In the event of a positive trial outcome, the intervention could next be translated to more premature infants and other minor painful skin-breaking procedures performed frequently in infants such as immunisation and cannulation and could be performed by parents or healthcare workers in the absence of parents. The Petal trial investigates a simple, free, low-risk, non-pharmacological pain-relieving intervention, which could be rapidly incorporated into routine clinical practice, benefiting infants, their parents and the wider community.

## Trial status

Participant recruitment is currently ongoing. Protocol version no. 3.0 (date of submission: 3 February 2022).

**Acknowledgements** The Authors thank the Wellcome trust and Bliss Charity for helping to fund the study, as well as the neonatal team, parents and babies who will help throughout the study.

**Contributors** MMC: conceptualisation, methodology, software, investigation, data curation, writing-original draft, writing-review and editing, visualisation, project administration. FM: conceptualisation, methodology, investigation, writing-original draft, writing-review and editing, funding acquisition. AGVH: methodology, investigation, data curation, writing-review and editing, project administration. DC: methodology, software, investigation, data curation, writing-review and editing. VM: conceptualisation, methodology, writing-original draft, project administration, funding acquisition. CH: software, methodology, data curation, funding acquisition. REF: conceptualisation, methodology, investigation, writing-original draft. SR: investigation. MvdV: software, validation, data curation, writing-review and editing. LB: methodology, data curation, writing-review and editing. EA: conceptualisation, methodology, writing-review and editing, supervision. RP: conceptualisation, methodology, investigation, data curation, supervision, project administration. AB: conceptualisation, methodology, supervision, writing-original draft, project administration, funding acquisition. RS: conceptualisation, methodology, resources, writing-original draft, writing-review and editing, supervision, project administration, funding acquisition.

**Funding** This work is supported by a Wellcome Senior Fellowship awarded to Rebeccah Slater (grant number 207457/Z/17/Z) and by Bliss (a UK charity) via a research grant (grant number N/A).

**Competing interests** None declared.

**Patient and public involvement** Patients and/or the public were involved in the design, or conduct, or reporting or dissemination plans of this research. Refer to the Methods section for further details.

**Patient consent for publication** Not applicable.

**Provenance and peer review** Not commissioned; externally peer reviewed.

**Data availability statement** Data sharing not applicable as no data sets generated and/or analysed for this study. Not applicable.

**ORCID iDs**

Annalisa G V Hauck http://orcid.org/0000-0002-6853-8363

Luke Baxter http://orcid.org/0000-0001-9548-7162

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
