## [Reviewer comments · BMJ Open]

ARTICLE DETAILS

TITLE (PROVISIONAL)	Study protocol: a multicentre, randomised controlled trial to investigate the effects of parental touch on relieving acute procedural pain in neonates (Petal)
AUTHORS	Cobo, Maria; Moultrie, Fiona; Hauck, Annalisa; Crankshaw, Daniel; Monk, Vaneesha; Hartley, Caroline; Evans Fry, Ria; Robinson, Shellie; van der Vaart, Marianne; Baxter, Luke; Adams, Eleri; Poorun, Ravi; Bhatt, Aomesh; Slater, Rebecca

VERSION 1 – REVIEW

REVIEWER	Colette Balice-Bourgeois Pediatric Institute of Southern Switzerland
REVIEW RETURNED	02-Mar-2022

GENERAL COMMENTS	Dear authors, Thank you for the opportunity to review the study protocol entitled "a multicentre, randomised controlled trial to investigate the effects of parental touch on relieving acute procedural pain in neonates (Petal)". This is a very important and timely study to determine whether a non-pharmacological intervention such as parental touch is effective in reducing procedural pain in neonates. Furthermore, it has the advantage of including parents in the care of their child as well as in the research (PPI). This protocol has many relevant literature references and in my opinion the rationale is comprehensive and explains all objectives, outcomes and methods very clearly. Although the study protocol is very well written, I would like to make the following suggestion: for the references in the text, some are before the point and some are after the point. For a better reading, I think it is better to always indicate them before the dot (or the comma). I hope that this research will bring new knowledge to be transferred into practice for the well-being of newborns and their parents.
---

REVIEWER	Xavier Durrmeyer Centre Hospitalier Intercommunal de Créteil
REVIEW RETURNED	04-Mar-2022

GENERAL COMMENTS	In this paper Maria Cobo et al. report a research protocol for an ongoing RCT to investigate the effects of parental touch on relieving acute procedural pain in neonates (PETAL). The topic is important, the question is relevant and the design is excellent. I only have minor comments and a few questions: - Rationale p. 3, l. 44-45: the benefits of maternal touch on growth and development are not based on a high level of evidence and I
--

	would suggest to tune down this statement;  - Intervention, p.6: my understanding is that no non-pharmacological pain control measure will be used (except parental touch if the study demonstrates its efficacy). Although the research team previously demonstrated the lack of efficacy of sucrose on noxious-evoked brain activity[1], sucrose and non-nutritive sucking are recommended by several academic societies for painful procedures, such as heel lance [2,3]. Why did the authors choose not to provide these interventions in both arms in addition to the assessed intervention? - Methods p.8: it is unclear for me if the PIPP-R Score will be rated by a single or multiple observers and if he/she/they will be blinded to the study arm. Finally, I have a few suggestions/questions regarding the analysis of future results:  - Is there a possibility that STAI scores or other outcomes - or even the intervention? - differ between fathers and mothers? This could be taken into account, in case randomization generates an imbalance in the type of parent who performs the intervention. - Is there a chance that the post-procedure parental touch will influence outcomes in the control group? Why not also perform a post-procedure touch in the intervention group? - Will the authors collect the number of previous painful procedures in infants, as they might influence the reaction to heel lance [4,5]? I realize that the comments related to methodology will probably not change the current design since the study is already recruiting, but it would satisfy my curiosity to have the authors' opinion on these points. Congratulations for conducting this important study and thank you for giving me the opportunity to review this manuscript. References  1 Slater R, Cornelissen L, Fabrizi L, Patten D, Yoxen J, Worley A, Boyd S, Meek J, Fitzgerald M: Oral sucrose as an analgesic drug for procedural pain in newborn infants: A randomised controlled trial. Lancet 2010;376:1225-1232. 2 Committee On Fetus Newborn Section On Anesthesiology Pain Medicine: Prevention and management of procedural pain in the neonate: An update. Pediatrics 2016;137:1-13. 3 Lago P, Garetti E, Merazzi D, Pieragostini L, Ancora G, Pirelli A, Bellieni CV, Pain Study Group of the Italian Society of N: Guidelines for procedural pain in the newborn. Acta Paediatr 2009;98:932-939. 4 Bembich S, Marrazzo F, Barini A, Ravalico P, Cont G, Demarini S: The cortical response to a noxious procedure changes over time in preterm infants. Pain 2016;157:1979-1987. 5 Gokulu G, Bilgen H, Ozdemir H, Sarioz A, Memisoglu A, Gucuyener K, Ozek E: Comparative heel stick study showed that newborn infants who had undergone repeated painful procedures showed increased short-term pain responses. Acta Paediatr 2016;105:e520-e525.
--	---

VERSION 1 – AUTHOR RESPONSE

Reviewer: 1

Dr. Colette Balice-Bourgeois, Pediatric Institute of Southern Switzerland

Comments to the Author:

Dear authors,

Thank you for the opportunity to review the study protocol entitled "a multicentre, randomised controlled trial to investigate the effects of parental touch on relieving acute procedural pain in neonates (Petal)". This is a very important and timely study to determine whether a non-pharmacological intervention such as parental touch is effective in reducing procedural pain in neonates. Furthermore, it has the advantage of including parents in the care of their child as well as in the research (PPI). This protocol has many relevant literature references and in my opinion the rationale is comprehensive and explains all objectives, outcomes and methods very clearly. Although the study protocol is very well written, I would like to make the following suggestion: for the references in the text, some are before the point and some are after the point. For a better reading, I think it is better to always indicate them before the dot (or the comma). I hope that this research will bring new knowledge to be transferred into practice for the well-being of newborns and their parents.

We thank the reviewer for the helpful suggestions.

While we agree that moving the references in the text as the reviewer suggests could improve readability, BMJ open guidelines require the placement of references after punctuation. We have edited the references in the text to comply with the journal guidelines (<https://authors.bmj.com/writing-and-formatting/formatting-your-paper/#references>) but suggest that the copy editors may choose to edit as suggested at their discretion.

Reviewer: 2

Dr. Xavier Durrmeyer, Centre Hospitalier Intercommunal de Créteil

Comments to the Author:

In this paper Maria Cobo et al. report a research protocol for an ongoing RCT to investigate the effects of parental touch on relieving acute procedural pain in neonates (PETAL). The topic is important, the question is relevant and the design is excellent.

I only have minor comments and a few questions:

- Rationale p. 3, l. 44-45: the benefits of maternal touch on growth and development are not based on a high level of evidence and I would suggest to tune down this statement;

Thank you for your helpful comment and interest in this work. As suggested, we have toned down the previous statement in the Introduction related to the benefits of maternal touch on growth and development. In addition, we have updated the references.

“Maternal touch behaviours are instinctive, evolutionarily conserved amongst mammals (Hertenstein et al. 2006)(Hertenstein et al. 2006). Previous studies suggest there may also be a potential relationship between enhanced maternal touch and infant growth and development (Fitri et al. 2021; Mrljak et al. 2022).” (p. 3, lines 83–85)

- Intervention, p.6: my understanding is that no non-pharmacological pain control measure will be

used (except parental touch if the study demonstrates its efficacy). Although the research team previously demonstrated the lack of efficacy of sucrose on noxious-evoked brain activity[1], sucrose and non-nutritive sucking are recommended by several academic societies for painful procedures, such as heel lance [2,3]. Why did the authors choose not to provide these interventions in both arms in addition to the assessed intervention?

Standard neonatal care in accordance with the local practice guidelines were provided to all neonates during the heel lance procedures. This included providing swaddling and non-nutritive sucking to all neonates in either arm of the study. This was not clearly described in the Protocol and has now been clarified in the 'Intervention' section on page 6.

“During the study all neonates received comfort care in accordance with the local practice guidelines. These measures included swaddling the infants and providing non-nutritive sucking.” (p.7, lines 171–173)

- Methods p.8: it is unclear for me if the PIPP-R Score will be rated by a single or multiple observers and if he/she/they will be blinded to the study arm.

Thank you for raising this important point. We have amended the 'PIPP-R score' section to clarify this issue.

“For each participant, PIPP-R scores will be assessed by investigators blinded to the study arm. A second investigator (blinded to the trial arm) will re-calculate 20% of the PIPP-R scores to measure inter-rater reliability.” (p.8, lines 245–247)

Finally, I have a few suggestions/questions regarding the analysis of future results:

- Is there a possibility that STAI scores or other outcomes - or even the intervention? - differ between fathers and mothers? This could be taken into account, in case randomization generates an imbalance in the type of parent who performs the intervention.

We agree with the reviewer that the efficacy of the intervention may differ when performed by either the father or the mother, and consider this to be an important and interesting question. We are collecting data about which parent performs the stroking for each participant and intend to explore this outcome as a post-hoc aspect of the study. At this stage we did not choose to stratify the groups based on this factor, however we will report these findings and may use this information to help interpret our results.

- Is there a chance that the post-procedure parental touch will influence outcomes in the control group? Why not also perform a post-procedure touch in the intervention group?

We agree that the post-procedural parental touch may influence some of the outcome measures recorded in the control group. However, the study was primarily designed to assess whether noxious-

evoked behavioural, physiological and brain responses that occur within 30 seconds following the noxious procedure are influenced by parental stroking. As the post-procedural stroking will not have occurred at this stage, it will therefore not influence these outcome measures.

Nevertheless, as correctly identified by the reviewer, the post-procedural parental touch could impact the secondary outcome measures such as the STAI-S scores. We chose not to repeat the stroking post-procedure in the intervention group because it would lead to a between-group imbalance in the number of times the parents stroked their infants, which could also potentially influence the results.

On balance, we considered it most appropriate to have the same amount of stroking in each trial arm and only consider the timing of the intervention. With regards to the STAI-S scores we will therefore be looking at whether parental involvement providing pain relief *during* the intervention reduces anxiety as opposed to after the intervention.

- Will the authors collect the number of previous painful procedures in infants, as they might influence the reaction to heel lance [4,5]?

We agree with the reviewer that this data is very important. We are collecting the number of previous painful procedures for all trial participants (this includes the total number of acute tissue-damaging procedures that each participant has experienced, such heel lances, injections, and intravenous cannulation). These demographic details will be reported, and future exploratory analysis could consider the impact of these interventions on the subsequent pain responses.

I realize that the comments related to methodology will probably not change the current design since the study is already recruiting, but it would satisfy my curiosity to have the authors' opinion on these points.

Congratulations for conducting this important study and thank you for giving me the opportunity to review this manuscript.

We thank the reviewer for the insightful comments.

References

- 1 Slater R, Cornelissen L, Fabrizi L, Patten D, Yoxen J, Worley A, Boyd S, Meek J, Fitzgerald M: Oral sucrose as an analgesic drug for procedural pain in newborn infants: A randomised controlled trial. *Lancet* 2010;376:1225-1232.
- 2 Committee On Fetus Newborn Section On Anesthesiology Pain Medicine: Prevention and management of procedural pain in the neonate: An update. *Pediatrics* 2016;137:1-13.
- 3 Lago P, Garetti E, Merazzi D, Pieragostini L, Ancora G, Pirelli A, Bellieni CV, Pain Study Group of the Italian Society of N: Guidelines for procedural pain in the newborn. *Acta Paediatr* 2009;98:932-939.

4 Bembich S, Marrazzo F, Barini A, Ravalico P, Cont G, Demarini S: The cortical response to a noxious procedure changes over time in preterm infants. Pain 2016;157:1979-1987.

5 Gokulu G, Bilgen H, Ozdemir H, Sarioz A, Memisoglu A, Gucuyener K, Ozek E: Comparative heel stick study showed that newborn infants who had undergone repeated painful procedures showed increased short-term pain responses. Acta Paediatr 2016;105:e520-e525.

VERSION 2 – REVIEW

REVIEWER	Xavier Durrmeyer Centre Hospitalier Intercommunal de Créteil
REVIEW RETURNED	21-Jun-2022
GENERAL COMMENTS	All my previous comments have been addressed. Probably, the new sentence p.7 l. 171-173 should be changed from past to future. Congratulations !